# Olfactory Optogenetics: Light Illuminates the Chemical Sensing Mechanisms of Biological Olfactory Systems

**DOI:** 10.3390/bios11090309

**Published:** 2021-08-31

**Authors:** Ping Zhu, Yulan Tian, Yating Chen, Wei Chen, Ping Wang, Liping Du, Chunsheng Wu

**Affiliations:** 1Institute of Medical Engineering, Department of Biophysics, School of Basic Medical Sciences, Health Science Center, Xi’an Jiaotong University, Xi’an 710061, China; jewel121@stu.xjtu.edu.cn (P.Z.); cnyulantian@stu.xjtu.edu.cn (Y.T.); ytc20201011@stu.xjtu.edu.cn (Y.C.); weiwcchen@xjtu.edu.cn (W.C.); 2Key Laboratory of Environment and Genes Related to Diseases, Xi’an Jiaotong University, Ministry of Education of China, Xi’an 710061, China; 3Biosensor National Special Laboratory, Key Laboratory for Biomedical Engineering of Ministry of Education, Department of Biomedical Engineering, Zhejiang University, Hangzhou 310027, China; cnpwang@zju.edu.cn

**Keywords:** optogenetics, olfactory, chemical sensing, neuronal, light

## Abstract

The mammalian olfactory system has an amazing ability to distinguish thousands of odorant molecules at the trace level. Scientists have made great achievements on revealing the olfactory sensing mechanisms in decades; even though many issues need addressing. Optogenetics provides a novel technical approach to solve this dilemma by utilizing light to illuminate specific part of the olfactory system; which can be used in all corners of the olfactory system for revealing the olfactory mechanism. This article reviews the most recent advances in olfactory optogenetics devoted to elucidate the mechanisms of chemical sensing. It thus attempts to introduce olfactory optogenetics according to the structure of the olfactory system. It mainly includes the following aspects: the sensory input from the olfactory epithelium to the olfactory bulb; the influences of the olfactory bulb (OB) neuron activity patterns on olfactory perception; the regulation between the olfactory cortex and the olfactory bulb; and the neuromodulation participating in odor coding by dominating the olfactory bulb. Finally; current challenges and future development trends of olfactory optogenetics are proposed and discussed.

## 1. Introduction

As one of the oldest sensory systems, the mammalian olfactory system is capable of recognizing thousands of different odorant molecules, which can help creatures avoiding danger, looking for food, identifying spouses. The olfactory system has evolved a mature and perfect odor information processing mechanism. Odorant molecules are firstly sensed by olfactory sensory neurons (OSNs) of the olfactory epithelium (OE), where odorant receptors (ORs) are expressed in OSNs and interact specifically with odorant molecules. An OSN expresses only one type of OR protein, which belongs to the superfamily of G protein-coupled receptors (GPCRs). The specific interactions of ORs and odorant molecules cause OSNs to generate an electrical signal, which can be transmitted to the glomerular layer of the OB. The odor information is then projected by the mitral/cluster (M/T) cells of the OB through the lateral olfactory tract into the olfactory cortex (OC), including the anterior olfactory nucleus (AON), the piriform cortex (PC), the amygdaloid cortex (AOC), the olfactory tubercle (OT), and the lateral entorhinal cortex (LEC) [1]. At present, there is a general understanding of the structure of the olfactory system. However, the detailed mechanism of olfactory system requires further exploration. Much progress has been made in exploring the olfactory information processing mechanism, which mainly focuses on the structure and function of ORs [2,3,4], internal neural circuits of the OB [5,6,7], and feedback and centrifugal modulation of the OB [8,9,10]. The research progress of olfactory coding has been reviewed by many excellent articles [11,12]. OB transmits sensory input from OE to the OC and is modulated by intrabulbar circuits and centrifugal inputs. Therefore, it is a very necessary and difficult task to understand how different circuits mediate the various aspects of odor information encoding in OB.

How to deliver accurate odor stimuli to sensory cells has been a key technical challenge for investigating the olfactory system for decades. The emergence of optogenetic technology provides a novel and promising tool for this challenge [13,14]. The principle of optogenetics is to use light to control genetically engineered neuron populations with millisecond precision to activate or silence cells. The most commonly used optogenetics probes include depolarization and hyperpolarization genetics tools, such as channelrhodopsin-2 (ChR2) and *Natromonas pharaonis* halorhodopsin (NpHR) [15,16]. Some more flexible and refined tools have been expanded, such as ChETA [17] and ReaChR [18], which can activate target neurons at a higher frequency. Optogenetics can be used to elucidate neural circuit activity by controlling specific neuronal populations, which has ushered in important breakthroughs in the field of neuroscience. At the same time, optogenetic technology has also been used to reveal the mysteries of the olfactory system. Some optogenetic transgenic animal models have been developed for the research on the chemical sensing mechanisms of biological olfactory system, and are illustrated in Figure 1. Light–sensitive proteins have been expressed in a variety of neurons in the olfactory system, including OSNs in the OE, main output neurons and inhibitory interneurons in the OB, neurons in the OC, and neurons in the neuromodulation system dominating the OB. Thanks to its relationship between mammals and humans, as well as its technological maturity and expansion, the applications of optogenetics in non-human primates have also progressed, focusing on the primary motor cortex (M1) [19] or the frontal eye field (FEF) [20].

Interestingly, the precise light stimulation can replace the unstable odor delivery when necessary, making optogenetic technology extremely attractive in olfaction research. In recent years, optogenetics has been extensively applied in many fields, and some excellent reviews have summarized the progress of the application of optogenetics in the research of hippocampus related to memory [21], the prefrontal cortex associated with cognition [22], the amygdala associated with pain and anxiety [23], and neurological disorders such as depression and psychosis [24]. However, although optogenetics has been widely applied in the research of the olfactory sensing mechanism, the important applications of optogenetics in the olfactory system have rarely been outlined and discussed. Grimaud et al., reviewed the contributions of optogenetics in the study of olfactory learning and memory, but in recent years, scientists have made many outstanding advances in olfactory research using optogenetics [25]. Here, we first introduce optogenetic tools in the olfactory system, and then mainly focus on the findings in OB that use optogenetics as a tool, including the transmission of sensory input, the perception of odor information, and functional neuronal circuits in OB. Lastly, the current challenges and future development trends of olfactory optogenetics will be proposed and discussed.

**Figure 1 biosensors-11-00309-f001:**
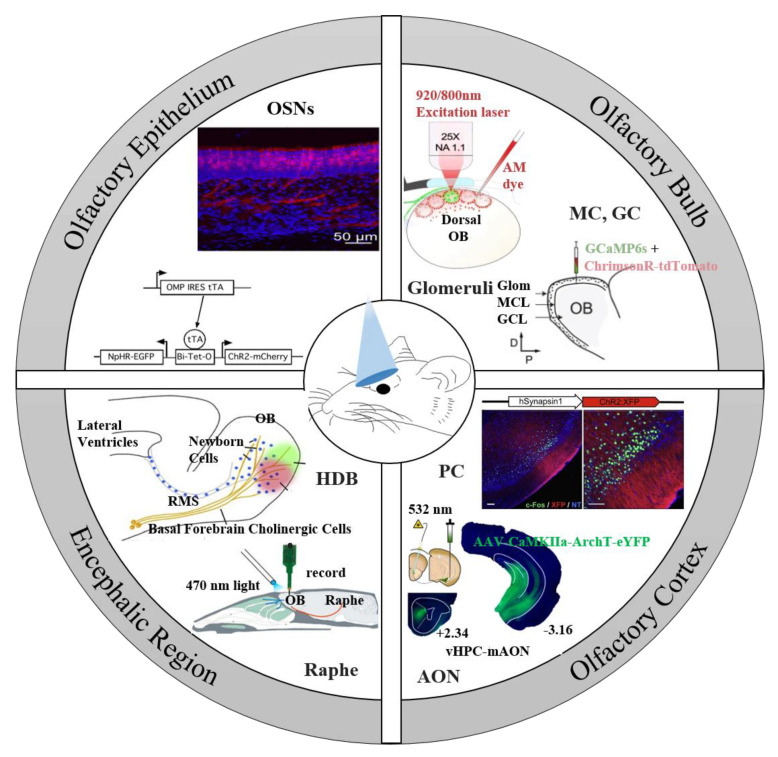
Application of optogenetics in various parts of the olfactory system. Olfactory epithelium: ChR2 is expressed in all OSNs to test whether mice can perceive the time of olfactory stimulation [26]. Olfactory bulb: ChR2 was expressed in a single glomerulus to study its response to optogenetic stimulation [27]. Express ChrimsonR-tdTomato in mitral cells and granule cells of mice to explore the coding characteristics of its perception detection [28]. Olfactory cortex: explore whether light-activated piriform cortex neurons of different subtypes can induce different behaviors [29]. Archaerhodopsin is used to inhibit the hippocampus dominating the AON subregions, revealing the principle of odor memory [30]. Encephalic region: ChR2 was expressed in the dorsal raphe nucleus to study the olfactory regulation of serotonin [31]. Express ChR2 in HDB and explore the olfactory perceptual learning involving cholinergic neurons [32]. Reproduced with permission from [26], Copyright 2012 Society for Neuroscience [27], Copyright 2018 Springer Nature [28], Copyright 2020 Elsevier [29], Copyright 2011 Elsevier [30], Copyright 2018 Springer Nature [31], Copyright 2016 Society for Neuroscience [32], Copyright 2019 Elsevier.

## 2. Optogenetic Tools for the Olfactory System

Optogenetic tools include dozens of light-sensitive proteins, which can be activated by different wavelengths of light with various operating speeds. Microbial rhodopsins undergo membrane depolarization or cellular signaling cascades caused by light-induced photochemical reactions [33], and are widely used in neuroscience, including the olfactory system. Channelrhodopsin-2 (ChR2) was the first optogenetic tool to activate neurons with light, and is the most commonly used one due to its rapid on-rate [16]. When ChR2-expressed neurons are illuminated by light with specific wavelength (450–490 nm), neurons will be depolarized by activating channels [14]. These light-sensitive proteins such as ChR2 could be introduced to olfactory neurons by a viral vector or transgenic animals. Under the driving of various promoters, adeno-associated viruses (AAVs) are used to target ChR2 in M/T cells, interneurons in OB, cells in AON, and PC [34,35,36,37]. Lentiviral vectors (LVs) are also used to target ChR2 in the rostral migratory stream (RMS) and PC [29,38]. Optogenetic tools can be guided to the nervous system through a viral expression system, and targeted neurons with the help of specific promoters are summarized in Table 1 and Table 2. Specific promoters drive light-sensitive proteins to target different neurons, such as Pcdh21 targeting M/T cells, TH targeting short-axon cells, CHR targeting interneurons in the EPL, and choline acetyltransferase promoter targeting choline acetyltransferase neurons [34,39,40,41]. This genetic technology enables specific types of neurons of the olfactory system to be activated or silenced by light in order to study the role and functional connection of specific neurons in the olfactory system.

Several transgenic animal lines have been created for the research of the olfactory system. Arenkiel et al., generated transgenic mice expressing ChR2-YFP from the Thy1 promoter for the precise and rapid activation of mitral cells in the OB [42]. Then, they used Thy1-ChR2-YFP mice for mitral-cell-specific light stimulation to map the functional connectivity between mitral cells and interneurons [40]. They also generated VGAT-ChR2 transgenic mice with ChR2-expressed GABAergic neurons and glycinergic inhibitory neurons to study the neuronal connectivity [43]. Additionally, Dhawale et al., generated transgenic mice by expressing ChR2-EYFP into OSNs and their axons with a promoter OMP [44]. In generated transgenic mice, the individual M/T cell can be activated by illuminating a single glomerulus. OMP-ChR2-YEP transgenic mice and M72-ChR2 transgenic mice lines were created to study olfactory perception (details in Section 4) [45,46]. In summary, these ChR2 transgenic animals allow scientists to selectively activate olfactory neurons and study the signal processing and perception in olfactory system [47]. 

Gunaydin and colleagues designed and verified the E123T mutation in ChR2 (ChETA) to address the precision limitations of ChR2 [48]. The Cre-dependent AAV-ChETA-EYFP vector was injected into the OT for cell-type-specific expression to study the roles of OT neurons in attractive and aversive behaviors [49]. Similarly, Aqrabawi and colleagues used AAV-ChETA-eYFP vectors to express ChETA-eYFP in AON neurons in FosCreER mice to manipulate populations of neurons in AON which constitute odor engrams [50]. Lin and colleagues engineered a variant of channelrhodopsins named denoted red-activatable ChR (ReaChR), which enables transcranial optical activation of neurons [51]. Using the novel optogenetic tool, Inagaki and colleague generated the UAS-ReaChR transgenic Drosophila for neural manipulation [52]. This transgenic model has helped to understand the neuronal functions of Drosophila mushroom body (a higher olfactory circuit) and the sensitivity regulation in insect primary olfactory neurons [53,54]. 

The proton pump archaerhodopsin (Arch) from *Halorubrum sodomense* or the archaerhodopsin from the Halorubrum strain TP009 (ArchT) were used as optogenetic neuronal silencing tools. To selectively silence GABAergic inhibitory neurons, scientists injected AAV-ArchT virus into granule cell layers in the OB of transgenic mice expressing Cre recombinase. This optogenetic technology, combined with intracellular recordings, examined the contribution of inhibition to rhythmic activity in the mouse olfactory bulb [55]. McCarthy and colleagues injected AAV-ArchT virus into the accessory OB of Pcdh21-Cre mice in which expression of Cre-recombinase is restricted to M/T cells to examine whether light inhibition of accessory OB neurons could affect lordosis in sexually mice [34]. In addition, another commonly used silencing tool is *Natronomonas pharaonis* halorhodopsin (NpHR). The modified NPHR is called eNpHR3.0, which has better targeting of cell membrane, a longer current, shorter response time, and more sensitive response. The pLenti-hSyn-eNpHR3.0-EYFP lentivirus [56] were injected in the OB to study the OB projection to the olfactory tubercle [57] and to examine the effect of anterior OB inhibition on odorant attraction [58].

**Table 1 biosensors-11-00309-t001:** Optogenetics approach in interneurons in OB.

Level	Expression Target	Model Animal	Expression Approach	Light Delivery					Electrophysiology Recordings	Behavior	Ref.
				Tool	Wavelength (nm)	Duration (ms)	Frequency (Hz)	Power (mW)			
GL	Glomeruli	OMP-ChR2-YFP transgenic mice	Transgenic animal model	A 470 nm LED coupled with an objective	470	10	-	-	M/T cells: patch-clamp	-	[6]
	SACs	TH-Cre mice	Injection of AAV-ChR2 into the GL	A solid-state laser coupled with an optical fiber	473	-	-	100	M/T cells or ETCs: whole-cell patch-clamp or cell-attached and tungsten microelectrodes	-	[39]
EPL	EPL-INs	Crh-Cre mice	Injection of AAV-ChR2 into the OB	A BLM-Series 473 nm blue laser system coupled with an objective	473			20–40	EPL INs: whole-cell patch-clamp	Olfactory associative learning training	[40]
	EPL-INs	CRH-Cre mice	Injection of AAV-ChR2 into the OB	A blue laser system guided by implanted fiber optics	473	10	-	30	MCs: whole-cell patch-clamp and extracellular recording electrodes	Olfactory associative learning training	[59]
IPL	dSACs	Chrna2-Cre mice	Injection of AAV-ChR2 into the IPL	A 75 W xenon arc lamp coupled with an objective	-	-	-	-	TCs: whole cell patch clamp	-	[35]
GCL	GCs	Dlx5/6-Cre mice	Injection of AAV-ChR2 into the OB	A BLM-Series 473 nm blue laser system coupled with an objective	473			20–40	GCs: whole-cell patch-clamp	Olfactory associative learning training	[40]
	GCs	OMP-Cre mice	Injection of AAV-ChR2 into the GCL	An implanted LEDs driven with a high-power LED driver	470	5	40	23	M/T cells: a silicon-based recording electrode and 32 channels optrode	Habituation task; Olfactory discrimination task	[60]

**Table 2 biosensors-11-00309-t002:** Optogenetics approach in neuromodulation projections to the OB.

Encephalic Region	Expression Target	Model Animal	Expression Approach	Light Delivery					Electrophysiology Recordings	Ref.
				Tool	Wavelength (nm)	Duration (ms)	Frequency (Hz)	Power (mW)		
Basal forebrain	HDB cholinergic neurons	VGLUT3-Cre mice	Injection of AAV-ChR2 into the HDB	A 75W xenon arc lamp coupled with an objective	-	10–20	-	-	OB cells: patch clamp	[61]
		ChAT-ChR2-EYFP transgenic mice	Transgenic animal model	A diode-pumped solid-state 473 nm laser coupled with an optical fiber target the HDB	473	15	5–50	-	M/T cells and brain slices: patch clamp	[41]
		ChAT-ChR2-EYFP mice transgenic	Transgenic animal model	A blue light diode laser and a blue LED coupled with implanted fiber	473	15	5–50	-	-	[32]
	HDB GABAergic neurons	DLX5/6-Cre mice	Injection of AAV-ChR2 into the HDB	A blueCoolLED pE 100 coupled with an objective	490	-	-	-	OB cells: whole-cell	[62]
		ChAT/GAD2-Cre mice	Injection of AAV-ChR2/eNpHR into the HDB	A 470 or 565 nm LED coupled with an optical fiber positioned in the OB	470; 565	10000	-	10;3	M/T cells: sixteen channel electrodes	[63]
Raphe nuclei	5-HT axons	TPH2-ChR2-YFP transgenic mice	Transgenic animal model	A bright light-emitting diode (LED) array coupled with a microscope	473	10	10	15	M/T cells: tungsten electrodes and whole-cell	[64]
	serotonergic cells	Slc6a4-Cre mice	Injection of AAV-ChR2 into DRN	A 470 nm LED coupled with a glass fiber positioned close to the OB	470	10000		1–10	OB cells: 16-channel electrode	[31]
	serotonergic cells	SERT-Cre mice	Injection of AAV-ChR2 in the DRN	A 470 nm laser coupled with an optrode lowered into the DRN	470	10	1–30	-	APC neurons: microelectrodes;	[65]
locus coreuleus	noradrenergic neurons	DBH-Cre-NpHR transgenic mice	Transgenic animal model	A solid-state laser coupled with an optical fiber implanted in the OB	532	-	-	2–10	MCs: tetrodes	[66]

## 3. The Sensory Input from OSNs to the OB

The OE, as the first level to receive odor information, plays an important role in the processing of odor information. OSNs expressing the same type of ORs usually project their axons into two bilateral glomeruli located in the OB [67,68]. When OSNs are stimulated by odors, the M/T cells in the OB will generate dynamic and rapid electrical signals [69]. Therefore, it is crucial to reveal how OSNs transmit odor information to M/T cells. Researchers expressed the opsins in OSNs of the OE and activated these neurons by light for the study of the light-induced responses of neurons in combination with electrophysiological, pharmacological, or behavioral methods [70,71]. The use of optogenetics has proven to be feasible to deliver precise stimuli in order to simulate the transmission of odor information from OE to OB as well as the processing process in OB [72]. 

Gire et al., used the patch-clamp to record light-induced OB signals in transgenic mice that expressed the ChR2 selectively in OSNs to investigate mechanisms of OSN signaling into mitral cells (MCs) [26]. Under the control of the OMP olfactory receptor promoter, ChR2-mCherry mice were crossed with mouse strains encoding the TTA gene to produce offspring that specifically expressed ChR2 in the OE. The patch-clamp recording of OB slices showed that most MCs did not display significant fast signals in response to light stimulation of ChR-expressing OSNs, and the OSN signals were shunted in MC. Recording the signals of tufted cells in the same way indicated that MCs received strong multistep signals through the tufted cells, suggesting that the olfactory information in OSNs was processed considerably before reaching MC, i.e., the tufted cells only mediated processing. The input from OSNs arrived at the glomerulus first, and each glomerulus accepted the input convergence of many OSNs expressing the same OR [73,74]. In addition, sister cells received input from the same glomerulus when the odor information in the OE reached the M/T cells from the glomerulus. Therefore, whether the higher olfactory center received redundant information was worth pondering. Albeanu et al., determined the M/T sister cells via the light-activated electrophysiological signals of a single glomerulus [44]. Transgenic mice expressing ChR2-EYFP in OSNs were used in this study to enable the glomerular layer (GL) to be accurately activated by light, while extracellular recordings showed that M/T cells only responded to the light stimulation of the GL. Because the size of the light spot was close to the average size of a single glomerulus, a single glomerulus can be activated for recording the response of M/T cells, which allowed the correct identification of the parental glomerulus to each mitral/tufted unit, and was further used as a basis to divide M/T cells into sister cells and non-sister cells. Studying the odor responses of sister cells found that the changes in their firing rates were related, indicating that sister cells received a common excitatory input. Moreover, the non-redundancy was found in the temporal characteristics of sister cell activity, which was of great significance for the transmission of information from the OB to the cerebral cortex.

In short, in the process of transmitting odor information to the OB, OSNs, glomeruli, MCs, and TCs played different roles to complete this task, respectively. After the direct OSN-EPSCs arrived at the TC, the MCs received a strong multi-step signal through the TC. Meanwhile, although a pair of sister cells (M/T) receive a common glomerular input, there are non-redundant odor responses. With the help of optogenetics, the goal of precise control of OSNs or a single glomerulus can be achieved, and then the downstream neural response can be measured to analyze the projection relationship with M/T cells.

## 4. How Do Activities of the Olfactory Bulb Neurons Affect Perception?

An important issue in olfactory research is how the brain encodes and processes odor information, i.e., how to convert chemical perception signals stimulated by odors into complex brain activities and behaviors. Imaging, electrophysiology, pharmacology, genetics, and other methods were used to reveal the basic principles of olfactory codes, including phase coding [75], combined coding [76], sparse coding [27], and concentration coding [77]. More studies on olfactory coding have been carefully discussed in some excellent reviews [11,78,79], and this review focuses more on the role of optogenetics in it. Light can selectively activate glomeruli instead of odor stimuli, and the perceptual responses can be explored by adjusting different characteristics of light stimulation, including the start and end time, the duration, the population, and number of glomeruli of light stimulation, etc. (Figure 2). As a result, odor perception is associated with complex neural activity patterns, and optogenetics is used as a powerful weapon to unlock the mystery of olfactory encoding.

The timing of odor stimulation is a key parameter and may represent important information of the olfactory system. In order to better understand the role of stimulus timing in the olfactory system, and to explore whether and how animals read the time of olfactory activation relative to the sniff cycle (‘sniff phase’), Rinberg et al., from the New York University Neuroscience Institute used optogenetics to generate time-controllable light stimulation to activate olfactory sensory neurons [45]. The results of behavioral and electrophysiological recordings indicated that the mouse’s olfactory system can accurately detect the beginning of the stimulus relative to the olfactory cycle and read out the time pattern. However, another group of researchers from the John B. Pierce Laboratory used the same method to prove that mice can successfully distinguish the onset time and delay of virtual odor stimulation, regardless of sniffing [80]. The opposite result did not negate the role of the sniffing cycle but indicated that the internal glomerular timing (the activation time relative to other neurons in the sequence) also participated in the encoding of time information, thereby independently affecting the perception of smell. Animals can not only perceive the beginning and end of light stimulus, but also distinguish the duration of the stimulus. Li et al., stimulated ChR2-expressed OSNs of mice at different durations [81]. Animal behavior results showed that mice could distinguish stimuli with a duration of 10 milliseconds, demonstrating that the mammalian olfactory system could accurately sense the difference in odor input duration. They also explored the thorny issue of neuron processing of input duration. The results of electrophysiological records showed that, when the glomeruli were activated by light of different durations, M/T cells responded strongly, and the frequency of their stimulation spikes carried the information of the stimulation duration. 

Using time-controllable precise light stimuli instead of odor stimuli, scientists discovered that the characteristics of the time activity can impact on olfactory perception, including the start time, end time, and duration of the stimulus. In addition, the neural activity that induces perception should also include some spatial characteristics, such as what cells are responding to and the impact of the response of a single cell or multiple cells on cognition. Rinberg’s group have made many striking breakthroughs on this subject using optogenetics technology in recent years. They first explored whether the mouse can perceive the activation of a single glomerulus [46]. The coding sequence of ChR2-YFP was inserted into the gene encoding the olfactory receptor M72 to activate individual glomeruli with light. By changing the intensity and duration of light stimulation and other parameters, it was found that a single glomerulus can convey odor information using intensity and time coding prompts. However, odor stimuli usually induced a group of glomerular responses, which not only varied with odor, but also varied with the concentration of a single odor. Their team proposed an odor coding scheme called “primary coding”, which meant that some of the earliest activated glomeruli were very crucial for identifying odors [82]. In order to test this hypothesis, they delivered light stimuli to the ChR2 expressing OSN to generate masking stimuli. Combined with the electrophysiological recording and behavioral tests, they found that the earliest induced neural activity can be used to make olfactory judgments, confirming this olfactory primary code scheme. 

Using optogenetic technology to explore the sense of smell, researchers can measure the limit of behavior discrimination by precisely manipulating a single feature (a single time or space feature), which confirmed the importance of the above single feature for olfactory perception. Thus, how will the combined characteristics of neuron spatiotemporal activities affect olfactory perception? Researchers performed optogenetic operations on mice to link the complex spatiotemporal activity patterns of neurons with olfactory perception. Chong et al., studied the importance of the combined characteristics of the spatial identity of glomerular activation and the temporal latency for perceptual meaning [83]. They used OMP-ChR2-YFP mice to manipulate the activated neuron groups or the activation latency with precise light stimulation, and then measured the changes in mouse cognition under different activity patterns. The results showed that activating different cell groups or changing their activation latency relative to other cells would cause varying degrees of changes in cognition. In this study, they did not explore the animal’s perception limit, and a more advanced optogenetic technology would provide the possibility to solve this problem. The holographic two-photon optogenetic technology can selectively activate neuron collections with the single cell resolution [84], achieve the spike–scale stimulation with millisecond resolution via the latest soma-covering light spots technology [85], and reproduce the temporal and spatial resolution of olfactory neuron activity patterns. In the latest study, Rinberg et al., used a modified light stimulation to activate neurons expressing red-shifted opsin ChrimsonR, and explored the three stimulus characteristic dimensions of mice (i.e., number of response cells, synchrony between neurons, and latency from inhalation onset) [28]. Their latest results showed that mice can detect a single action potential synchronously evoked by less than 20 M/T cells. At the same time, the synchronization of activation between neurons affected the olfactory perception of mice more than the inhalation period. 

Odor stimulation induces neuronal activity, and the spatio-temporal combination of glomerular activity corresponds to different odors and even their concentrations [86], so the temporal and spatial features of odor perception have extraordinary significance for odor decoding. Recent research has explored the influence of multi-dimensional features such as number, synchronization (relative to the inhalation phase or not), latency and duration of glomeruli, and M/T cells activation on olfactory perception. Dr. Rinberg’s group has made outstanding contributions to the research on this topic. Although it may not completely replace natural odor stimulation, optogenetics provides precise parameterization and good control of causal operation with minimal damage, combines behavioral means to explore the contribution of these features to olfactory perception, and helps us better understand the basic principle of olfactory coding.

## 5. The Function of OB Interneurons in Odor Information Processing

The OB is the first relay station for odor information processing, connecting the sensory input of the OE with the olfactory area of the brain. The M/T cells in the OB receive odor information from the glomerulus, and the information is then transmitted to higher brain regions for further processing. M/T cells are modulated by various local interneurons in the OB during odor information processing. As shown in Figure 3, modulation involved neurons mainly include external tufted cells (ETCs), periglomerular cells (PGCs), superficial short-axon cells (sSACs) in the glomerular layer (GL), external plexiform layer interneurons (EPL-INs), deep short-axon cells (dSACs) in the internal plexiform layer (IPL), and granule cells (GCs) in the granule cells layer (GCL). Understanding the basic function of neuronal circuits in the OB may help decrypt the complex odor coding principles. Recently, optogenetics has been employed to reveal the complex modulation mechanism in OB. Table 1 summarizes the application of optogenetics approaches in investing the function of OB interneurons. Similar to the section on the functional connection between OE and OB, the general strategy of optogenetics in OB interneurons is to manipulate the characteristics of neuronal activation (in this case, the type of cell) and then measure the response of the downstream neuron to explore the projection from one area to another, where it is mainly the regulation of various types of interneurons on the M/T cells of main cells in OB.

### 5.1. The Glomerular Layer

The strong inhibitory effect of the glomerular circuit on M/T cells has received widespread attention. Recent studies have explored the different inhibitory effects of the GL on two main neuronal M/T cells of the OB [6]. The GL of OMP-ChR2-YFP transgenic mice was briefly stimulated by light and the membrane potential of M/T cells in OB slices was recorded to observe the different responses of MC and TC to glomerular light stimulation. Combined with the electrical stimulation, it was found that MC and TC exhibited different time patterns of odor-induced responses due to the differences in the inhibition of glomerular layer-mediated blockade. Similarly, Liu et al., studied the potent inhibition of the interglomerular circuit (IGC) formed by short axon cells (SACs) on M/T cells (Figure 4a) [39]. They expressed ChR2 in SACs by injecting the Cre-inducible Adeno-associated virus serotype 9 (AAV2.9) carrying fusion genes for ChR2 into the GL of the OB, and then recorded the response of M/T cells to light-activated SACs using in vivo and in vitro electrophysiological techniques. The results indicated that the IGC suppressed MTC through parallel circuits. In order to activate the GL by light, transgenic, or direct injection methods can be used to express opsin in the GL. Due to the introduction of optogenetics, it has become possible to undertake an in-depth study on the inhibitory effect of the glomerular layer on OB output neurons.

### 5.2. The External Plexiform Layer

In addition to receiving input from OSNs via synapses in the glomeruli, M/T cells also receive input from some inhibitory interneurons in the EPL [87]. Recent studies have drawn functional connections between M/T cells and EPL interneuron populations to help understand the basic function of neuronal circuits (Figure 4b) [40]. Huang et al., injected a ChR2 gene-loaded virus into MCs to generate photoactivable MCs. Then, a whole-cell patch clamp was used to record the responses of interneurons in the EPL. Using this method, the connectivity map of MC to EPL neurons was successfully established. Similarly, they completed the functional mapping of the EPL interneurons to MCs connection. Through comparison, it was found that there are certain differences in the inhibition patterns of EPL interneurons and GCs on MCs. This discovery further deepened our understanding of the OB microcircuits. In particular, the researchers, by means of optogenetics and electrophysiological recording, discovered a subpopulation of previously uncharacterized corticotropin-releasing hormone (CRH)-expressing interneurons located in the EPL, which can strongly inhibit MCs firing [59].

### 5.3. The Internal Plexiform Layer 

Due to the low abundance of deep short-axon cells (dSACs) in the internal plexiform layer (IPL), they have not been systematically studied. Therefore, their roles in the olfactory information processing are still unclear. Previous studies have found that dSACs have a large number of axons that project into the neurons of GCL, EPL, and GL, thereby affecting their odor-induced activity [88]. It has been recently found that the activity of glomerular layer-projecting deep short-axon cells (GL-dSACs) may be closely related to the balance of activities between MCs and tufted cells (TCs) (Figure 4c) [35]. Researchers observed the distribution of GL-dSACs in MOB, IPL, and superficial GCL using chrna2 labels. Then, the stable neuron firing was recorded using a whole-cell patch clamp. AAV was injected into MOB to express ChR2: mCherry in GL-dSACs. By activating GL-dSACs by light, it was found that GL-dSACs targeted PGCs, ETCs, and TCs in a single frequency, reflecting the regulation of GL-dSACs on MOB circuits. In addition, GCL- and EPL-projecting dSACs could innervate many granule cells and may have an inhibitory effect on M/T cells [89].

### 5.4. The Granule Cells Layer

Granule cells (GC), as GABAergic neurons in MOB, are considered to be the most abundant interneurons and the source of most M/T cell inhibition. Previous studies have reported that OB model separation plays a key role in olfactory learning [60]. In one study, ChR2 was expressed into the interneurons of the granular cell layer (GCL) by injecting AAV. It was found that light activated GCs significantly inhibited the odor response of M/T cells and promoted pattern decorrelation. Combining behavior methods and pharmacology, it was found that interneurons in the GCL involved in modulating the decorrelation of M/T cell activity patterns. In addition to studying the feedback inhibition of M/T cells by GCs, researchers also paid attention to how GCs were activated by the sensory input [90]. Optogenetic methods was also used to explore the transmission of information from OSNs to GCs and feedback inhibition from GCs to M/T cells. OMP–ChR2: EYFP transgenic mice were used to express ChR2 in olfactory sensory neurons, and patch clamp was used to record the firing activity of CGs and dSACs in the GCL. The activation of glomeruli induced the feed forward inhibition of GCs originating from dSACs, which is considered to be an important pattern of GC inhibition activity together with the asynchronous excitation of GCs by synapses. 

### 5.5. The Rostral Migratory Stream

The OB interneurons are produced in the subventricular zone and migrate along the rostral migratory stream (RMS) to the OB, where they differentiate into two main types of interneurons: granule cells and periglomerular cells (PGCs). The OB recruits thousands of adult-born neurons every day; so how do these adult-born neurons integrate into the adult neural network, what roles do they play in the olfactory bulb, and how do they affect the sense of smell? In order to understand these problems, scientists express opsins in RMS, activate adult-born neurons, and then measure downstream cell response or olfactory discrimination. In addition, there are some researchers using light-activated RMS to study the migration mechanism of adult-born neurons [91], but it does not involve olfactory processing etc., so it is not our focus.

Dr. Lledo’s team from Institut Pasteur has done a lot of excellent work on this topic. In order to understand how the newborn neurons in the olfactory bulb integrate into the pre-existing circuit, a lentiviral vector encoding ChR2-YFP was injected into the RMS of mice (Figure 4d) [92], and blue light was used to control the firing pattern of the newborn neurons. The results of patch clamp recording showed that all major cell types in the olfactory bulb received GABAergic synaptic output from adult-born neurons, and the signal characteristics of this output did not change over time. Another work specifically focused on granular cells born at different stages and examined the difference in synaptic transmission between “postnatal-born” granule cells and adult-born granule cells [93]. The study controlled the transmitter release of the two GC populations with light, and the IPSC of the downstream MC were recorded by the patch clamp. The results showed that, unlike “postnatal-born” GC, adult-born granule cells can resist the GABABR depression of GABA release, resulting in unique effects on olfactory behavior. So how does the GABA inhibition of these newborn neurons affect the function of the olfactory circuit? They then carried out this further work [38]. Similar to the previously mentioned strategy, the firing activity of adult-born neurons was controlled by the light stimulation, and then the mice’s odor discrimination ability and learning ability and MC firing activity were examined. The results of this detailed work showed that the light activation of adult-born neurons can directly affect odor learning and memory, and that a specific frequency of light activation inhibited the spontaneous odor-induced cell firing of MC. Next, they explored the connection between adult-born GC and odor coding [94]. They designed clever behavioral experiments to train mice to recognize rewarding light stimuli and measured the efficiency of goal-directed behavior, confirming that adult-born neurons were involved in the odor–reward connection. 

In earlier studies, anti-mitotic drugs [95] and transgenic ablation [96] were used to eliminate or reduce the proliferation of adult-born neurons in order to assess the functional contribution of these new neurons. However, it did not completely selectively induce the ablation of adult-born neurons, so it was difficult to determine the functional role. Using optogenetic methods, the synaptic connections between adult neurons and OB cells can be directly identified with the aid of behavioral and electrophysiological recording tools, as well as the effects of the activation of these cells on olfactory learning and memory. As mentioned above, adult-born neurons differentiate into GCs and PGCs in OB, and the influence of PGCs born in adulthood on olfactory behavior remains to be clarified. 

The OB acts as a relay between the olfactory epithelial sensory input and the advanced olfactory brain area. A large number of neural activities related to olfactory information processing are compressed and completed in the OB. Optogenetics provides a bridge to explore a variety of interneurons and complex circuits in the OB. In the glomerular layer, PGCs mediate the difference in inhibition of MCs and TCs, resulting in different spike latency of the two output neurons, and the inhibition mediated by superficial short-axon cells can reduce the response of M/T cells to the sensory input. The EPL interneurons are widely connected with MC, and one of the interneuron subgroups inhibits the activity of MCs while receiving excitatory input from MCs. A small number of interneurons in the IPL are also involved in the regulation of the OB output neurons. The microcircuit formed by the most abundant granule cells in the OB and MT cells plays a unique role in information processing and participates in pattern separation and odor recognition through this interconnection. Reviewing these studies, it was found that various types of interneurons jointly modulate the activity of the main output neurons in the OB through different methods of inhibition to realize the fine processing of olfactory information, and the role of optogenetics was throughout.

**Figure 4 biosensors-11-00309-f004:**
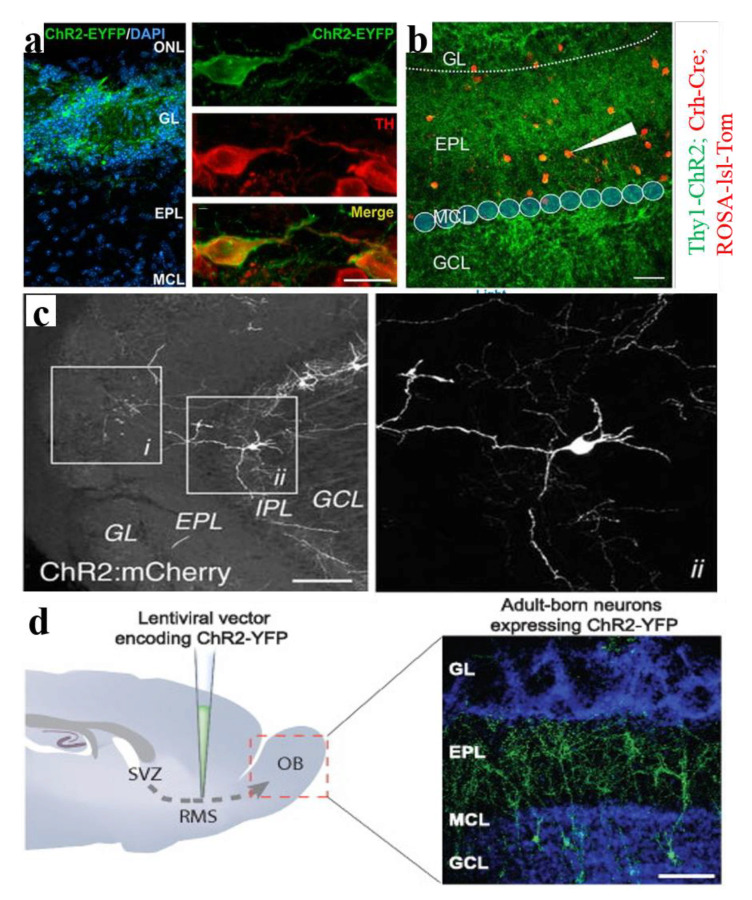
Cell-type-selective investigation of MOB inhibitory circuits. (**a**) The selective expression of ChR2-EYFP in the GL and brief optical stimulation of SACs inhibits MTC firing [39]. (**b**) Diagram of the MC-to-EPL interneuron connectivity mapping [40]. (**c**) ChR2:mCherry expression in the IPL and representative cell-attached recording of a GL-dSAC after brief photostimulation [35]. (**d**) The immunoreactivity of ChR2-YFP in adult-born neurons (green) and DAPI staining in all cell nuclei (blue), which can be activated by flashes of blue light (∼470 nm) [92]. Reproduced with permission from (**a**) [42], Copyright 2016 Society for Neuroscience; (**b**) [43], Copyright 2016 Society for Neuroscience; (**c**) [37], Copyright 2016 Society for Neuroscience; (**d**) [92], Copyright 2010 Society for Neuroscience.

## 6. Cortical Projection and Centrifugal Regulation

M/T cells transmit the odor information from the OB to the OC area through the lateral olfactory tract (LOT), where the odor information is integrated. How the cerebral cortex network recognizes and responds to these different synaptic inputs is an important topic. In order to clarify these mechanisms, researchers use optogenetics to study how the projection of OB to OC and the feedback projection affects the OB circuit, thereby participating in olfactory coding. The related olfactory cortex includes the anterior and posterior piriform cortices (aPC and pPC), the anterior olfactory nucleus (AON), the olfactory tubercle, and other related regions.

The piriform cortex (PCx), the largest olfactory area with a simple three-layer structure, participate in the recognition and memory of smell [97], which receives excitatory input from MT cells. Using ChR2 to lightly stimulate the glomeruli, it was found that the piriform cortex integrates and converges the excitatory input from OB [42]. Thus, how do the downstream neurons in the cortex decipher the spatiotemporal activities from the olfactory bulb that carry odor information? Using optogenetic manipulation of the spatiotemporal activity of olfactory bulb neurons in mice (the relative timing of the activation of the two spots), the firing activity of piriform cortical neurons was subsequently measured, instead of measuring the animal’s olfactory discrimination behavior mentioned above [98]. The results indicated that the olfactory cortex used the firing rate of cortical neurons to characterize the relative time code of peripheral olfactory stimulation. For the intensity of the odor stimulus, the characterization of PCx used a multiplexing strategy (activating neuron identity and overall synchronization) [99]. PCx relied on the concentration-invariant mechanism to stably characterize odor recognition at various concentrations [100]. That is to say, PCx only responds to the earliest activated OB input, and ignores the influence of the less selective input that arrives later, which can be further explored through adjusting the intensity of light stimulation to manipulate the number of activated neurons [8].

Pyramidal cells of PCx receive sensory input from OB, and ipsilateral OB receives centrifugal input from PCx. This cortical feedback directly stimulates GABAergic GCs and short axon cells, thereby inhibiting OB output neurons [37,101]. The researchers selectively activated ChR2-expressing cortical fibers, and then measured the firing activities of M/T cells, GCs, short axon cells, and periglomerular cells in the OB in order to determine the projection target from the cortex to the OB. In fact, with the deepening of olfactory research, the piriform cortex has attracted more and more attention. These problems involve intracortical connections [36], afferent circuits [102], interneuron suppression circuits [103], and plasticity associated with odor recognition, and finally decrypt the coding mechanism in piriform cortex. We recommend to readers the excellent reviews of Blazing et al., to understand these issues in detail [97,104]. Optogenetics illuminates the functional neural circuit of the projection of OB to PCx and feedback regulation from PCx to OB, which benefits from the inherent advantages of optogenetics in examining the connectivity between different regions.

The axons of the pyramidal cells in piriform cortex project to the ipsilateral OB of the cortical projection, while the neurons in the anterior olfactory nucleus project axons to the ipsilateral and contralateral OB. The cortical input of the OB is believed to mainly dominate the granule cell layer [105,106], and AON may also directly stimulate M/T cells [107]. The regulation and influence of this cortical feedback needs further examination. Recent reports have used optogenetics to study how these feedback projections affect the OB circuit [37,108]. ChR2-expressed AON was stimulated by blue light to record the light-evoked responses in the OB to check their functional synapse connections using in vivo and in vitro electrophysiological techniques. They found that activation of the AON can inhibit MC, which is mediated by GCs and interneurons in the glomerular layer. The AON and aPC have been reported to perform similar functions in the effect on odor-evoked responses of M/T cells [37,109]. Mazo et al., investigated the AON and aPC as a single functional entity and examined the mechanism in which the OB receives top-down input from the OC (Figure 5c) [108]. By expressing ChR2 in the AON and aPC, GABA intermediate neurons can be activated by light. It was found that the activation of intermediate neurons could inhibit the synaptic connections between the AON and aPC to GCs, thus weakening the inhibitory effect on M/T cell activity. These studies further revealed the top-down modulation mechanism of the cerebral cortex on the OB. Another research team from Institut Pasteur investigated an interesting aspect of the OC [110]. To identify the origin of the presynaptic inputs to new GCs, ChR2-eYFP was expressed in the OB, and ChR2-mCherry was expressed in the AON. Using this labeling scheme, it was observed that the cortex is the main source of excitatory input of the GCL, which could be strengthened by learning. In conclusion, the top-down input from the AON in the OB is mainly targeted at granule cells and modulates OB output through GC inhibition, while GABAergic interneurons in the AON can modulate such projections. Through this complex mechanism, the effect of the cortex on the OB is precisely refined.

The use of optogenetics for investigating the OC mainly involves the aforementioned parts: the piriform cortex and the anterior olfactory nucleus. In addition to the olfactory tubercle, the periamygdaloid cortex and the anterior part of the entorhinal cortex are also considered to be main parts of the OC. The extensive olfactory network also includes the orbitofrontal cortex, thalamus, and insular cortex [111]. While there have not been enough systematic studies to fully understand this area, a few studies have employed optogenetics to glance at the working mechanism of these mysterious cortex. The lateral entorhinal cortex (LEC) directly receives input from the OB as the olfactory limbic circuit. This input and the effect of OB activity on the limbic circuit are not clearly recognized. Gretenkord et al., studied the structure and functional principles of communication between the OB and the lateral entorhinal cortex (LEC) [112] by injecting AAV-ChR2-EYFP into the OB and activating M/T cells with blue light. Then, the in vivo electrophysiological signals of the LEC indicated that the activation of M/T triggered theta oscillation in LEC. These seldom mentioned cortical areas clearly also participate in olfactory information processing in different ways.

The olfactory cortex is involved in all aspects of odor perception, including odor intensity, odor recognition, and odor memory [113]. The cortex not only receives sensory input from OB, but there is also a projection from one area to another of the cortex. For example, AON and basolateral amygdala (BLA) are involved in regulating the activity of PCx odor activity (Figure 5b) [47,114,115], while aPC and prelimbic prefrontal cortex (plPFC) both receive direct projections from AON [114,116]. The activation or inhibition of axons from the medial prefrontal cortex (mPFC) to the anterior agranular insular cortex (aAIC) affects working memory [117]. In addition, OC also receives other neuromodulation inputs [118]. The focus of our review is the functional neuron circuit between the cortex and the olfactory bulb. Optogenetics can selectively activate the olfactory bulb or cortical neurons, and even axon fibers, illuminating the cortical projection and cortical feedback of the OB.

**Figure 5 biosensors-11-00309-f005:**
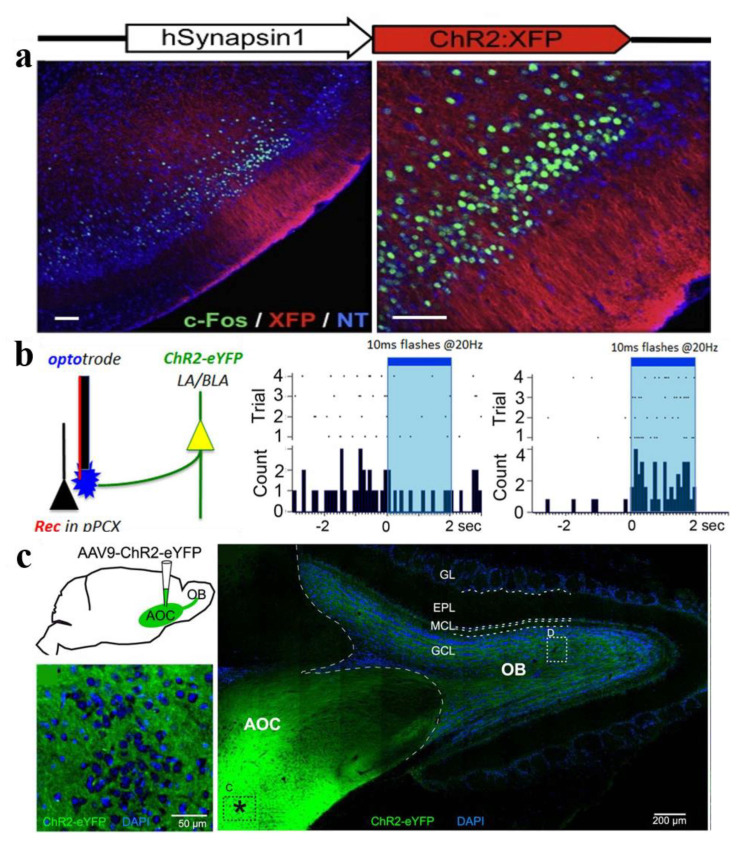
Optogenetic approaches in OC. (**a**) Expression of ChR2 in layers two and three of piriform cortex (PC) [29]. (**b**) Diagram indicating the infected LA/BLA pyramidal cell populations that extend axons into the PCX [114]. (**c**) Schematic representation of AAV2/9-ChR2-eYFP injection into the AOC and cortico-bulbar ChR2 axons targeting the OB and the expression of ChR2 in the AOC [108]. Reproduced with permission from (**a**) [29], Copyright 2011 Elsevier; (**b**) [117], Copyright 2014 LaLuniere; (**c**) [111], Copyright 2016 Society for Neuroscience.

## 7. The Neuromodulation in Odor Coding by Dominating the OB

In addition to receiving centrifugal feedback regulation from OC, OB is also regulated by the neuromodulation system, including cholinergic [119,120] and GABAergic [121] neurons in the basal forebrain (BF), serotonin (5-hydroxytriptamine; 5-HT) neurons in the dorsal raphe nucleus (DRN) [64,122], and noradrenergic (NA) neurons in the locus coeruleus (LC) [123]. These projections are mainly focused on the GL and IPL in the OB, which are involved in odor detection and recognition and olfactory learning. Table 2 summarizes the application of optogenetic approaches in neuromodulation.

### 7.1. The Horizontal Diagonal Band of Broca

The OB receives centrifugal input from both cholinergic and GABA neurons in the horizontal diagonal band of Broca (HDB), one BF nucleus. As such, light stimulation is needed to selectively activate target neurons and study their respective mechanisms of modulation. 

In order to examine the special functions of cholinergic neurons projecting to the OB, many in-depth studies have been carried with the help of optogenetic tools. Recently, by injecting AAV-ChR2-EYEP into the HDB to selectively activate the projection of cholinergic neurons expressing vesicular glutamate transporter 3 (VGLUT3) to the OB, Case et al., investigated the modulation effect of this subset of BF neurons on the OB circuit [61]. Fluorescence imaging of acute slices showed that the projection of cholinergic neurons to the OB was enriched in the IPL. Combining light stimulation and whole-cell patch clamp technology, the results indicate that activation of the HDB may strongly affect the activity of deep short-axon cells (dSACs) in the IPL. Similarly, Ma et al., used the same method combined with in vivo electrophysiological recordings and found that light stimulation of HDB cholinergic neurons can inhibit the spontaneous firing activities of M/T cells, periglomerular cells, and granule cells in the OB (Figure 6a) [41]. In addition, cholinergic neurons in the HDB are involved in the manipulation of olfactory perception learning (OPL) [32]. Combined with behavioral studies, light-activated cholinergic projection in the HDB expressing ChR2 enhanced the olfactory discrimination ability of mice. In summary, cholinergic neurons in the HDB regulate various interneurons and M/T cells in the OB by releasing acetylcholine, thereby regulating olfactory information processing and olfactory learning.

Another centrifugal projection of the basal forebrain to the OB comes from GABAergic neurons in the HDB. Yet, GABAergic BF-OB projections have received relatively little attention. Diez et al., studied the target neurons of GABAergic BF-OB projections [62] by injecting ChR2-expressing virus into the HDB of dlx5/6-Cre mice, which makes the GABAergic axons of the basal forebrain photoactivatable. Whole-cell recording results showed that light-activated GABAergic neuron projections can innervate specific subtypes of periglomerular cells and extensively regulate the OB circuit. Another study employed optogenetic methods to investigate the manipulatory effect of GABAergic BF-OB projections on M/T cells (Figure 6b) [63]. The results revealed an interesting fact that the activation of GABAergic projections from the basal forebrain to the OB can inhibit the spontaneous firing activity of M/T cells but excite their odor-induced firing activity. This suggests that GABAergic BF modulation plays an important role in the OB circuit. Clarifying the modulation of the basal forebrain on the OB may help us understand the processing of complex olfactory information.

### 7.2. 5-Hydroxytriptamine

5-Hydroxytriptamine (5-HT), a neurotransmitter released by neurons in raphe nuclei, is related to various brain functions, including sensory processing in the olfactory system. Optogenetic tools have played an irreplaceable role in studying how 5-HT affects OB odor information processing. Researchers injected virus expressing ChR2 to the raphe nuclei to study the mechanisms of 5-HT OB projections [64]. The results of the whole-cell patch-clamp recording showed that the brief light activation of the raphe nuclei can quickly excite the spontaneous firing of M/T cells, generally enhance the odor-induced activity of TCs, and modulate the odor-induced activity of MCs, bidirectionally. This kind of various modulation pattern of the raphe nuclei on M/T cells can regulate the OB output and improve the mode separation of odor. In addition to the output neurons in OB (M/T cells), serotonergic projections from the raphe nuclei are also involved in the regulation of OB interneurons, such as PG cells and short-axon cells [31]. The results of extracellular recordings indicate that light-activated serotonergic projections from the raphe nuclei to the OB can significantly enhance the responses of interneurons (e.g., PG cells and SACs) to sensory input.

Another interesting study investigated the regulation of 5-HT activation on the OC of mice, specifically the anterior piriform cortex (aPC) (Figure 6c) [65]. Researchers injected ChR2-expressing Adeno-associated virus into the brainstem dorsal raphe nucleus (DRN) and observed the spontaneous firing of neurons in the aPC after photoactivation, further verifying the role of 5-HT in sensory processing.

### 7.3. Noradrenergic (NA) Neurons in the Locus Coeruleus 

Norepinephrine from the locus coeruleus (LC) plays a key role in olfactory perception learning. A combination of optogenetic and behavioral studies in mice investigated the value of norepinephrine in olfactory learning tasks by representing odors through OB oscillations (Figure 6d) [66]. The expression of NpHR in LC-NA axons under a specific promoter enables the axons to be silenced by light stimulation. An optetrode was used for electrophysiological recording from the OB. After mice learned to distinguish odors through behavioral training, researchers observed that the light silencing of LC-NA axons changed the ability of mice to distinguish smells by oscillating the OB.

Neurotransmitters play an important role in the signal processing of the OB. Together with the neurons in the OC, these modulatory neurons input a large amount of centrifugal feedback to the OB [124]. They participate in olfactory processing by regulating output neurons and a variety of inhibitory interneurons in the OB or participate in olfactory perception learning by affecting the cortex. 

Studies revealed the modulating effect of neurotransmitters on the olfactory bulb with the help of optogenetics. Firstly, the photogenic activation of HDB cholinergic neurons increased the spiking response of M/T cells to odors, suggesting cholinergic neurons improve the possibility of olfactory recognition and participation in olfactory perception learning. Different studies have observed inconsistent cholinergic stimulation to enhance [63] or inhibit [41] the activity of M/T cells in the resting state, which may be due to the difference in optogenetic stimulation of cholinergic axons to the OB and cholinergic neurons. GABAergic centrifugal projections from BF onto all layers of the OB inhibit the spontaneous firing of M/T cells, enhance the odor-evoked activity of M/T cells, and extensively regulate the olfactory bulb circuit. The co-transmission of cholinergic and GABAergic neuron projections onto the axon of OB indicates the complex regulation of BF neuron projections to MOB target neurons. Secondly, light activation of 5-HT excites the spontaneous firing of M/T cells, enhances the odor-evoked response of TC, and regulates the decorrelation of MC odor response. Additionally, light activation of 5-HT strongly enhances the spontaneous firing of PG cells, SACs, and other interneurons, and moderately enhances their odor response, while the activation of DRN 5-HT neurons inhibits the spontaneous firing of neurons in the olfactory cortex. Finally, the optogenetic silencing of noradrenergic neurons changes the odor-induced local field potential oscillations. In short, neurotransmitters from different encephalic regions have more complex and diverse regulatory mechanisms for MOB than expected. Some studies have revealed their possibility of improving olfactory recognition, which provides a clue for the development in biosensing and other fields.

## 8. Conclusions and Perspectives

The mammalian olfactory system has a powerful ability to process olfactory information, including the ability to recognize and distinguish thousands of odors, distinguish different intensities of odor stimuli, and even perceive the difference in the duration of odor stimuli in milliseconds, all of which requires the orderly coordination between different parts of the olfactory system. Therefore, decrypting the mechanism of the functional connections in the olfactory system is the focus of olfactory research. Optogenetics has become one of the most suitable and powerful tools for olfactory research. The general strategy of this method is to activate or silence target neurons in the olfactory system through light stimulation, and then combine electrophysiology to measure downstream neuron responses, or combine behavior to measure animal olfactory perception, to study the projection from one area to another and the impact of the target area on olfactory perception. Odor information is directly input from the OSNs in OE to OB, adjusted by the inhibitory circuit and centrifugal modulation, and projected into the cortex for characterization. Such a complex process involves the participation of many neurons, and optogenetics has played an instrumental role in illuminating these pathways. 

Some key issues need to be focused on in the application of optogenetic technology to study the olfactory system. The first is the choice of optical probes, which are used to activate or silence olfactory neurons according to needs. As we can see, a wild variety of virus vectors and transgenic animal models allows scientists to target ChR2 to almost all neurons in the olfactory system, including OSNs in the OE, M/T cells, and inhibitory neurons in the OB, and neurons in the OC. The mutations of ChR2 (ChETA) provide higher precision control and faster spiking frequency, and may be more suitable for research requiring accurate spike times. Transgenic Drosophila models of ChR2 and ReaChR are both broadly applied in studying the insect olfactory system, while some scientists believe that ReaChR works better in adult Drosophila owing. Interestingly, the application of optogenetic silencing tools is far less extensive than the activation tools in the olfactory research, considering that silencing neurons is harder to trace its origin than exciting neurons, but there is still a certain demand. Optogenetic neuronal silencing tools (ArchT and NpHR) have been used to target M/T cells and inhibitory neurons in the OB by virus vectors. Despite the development and progress of the ChR2 targeting strategies for olfactory neurons, other optogenetic tools and efficient targeting strategies remain to be developed. For instance, the mutations of ChR excel ChR2 in precision control and spiking frequency, so these are needs for generating transgenic animal models to target these tools to various neurons in the olfactory system. Similarly, the silencing tools needs potent target strategies for not only neurons in the OB but also neurons in the OE and the OC. In addition to eNpHR3.0, some other excellent optogenetic silencing tools with enhanced functions have been generated, which have not been used in the olfactory study due to the lack of cell type-specific promotors and transgenic animal models. Jaws derived from *Haloarcula (Halobacterium) salinarum* exhibit robust inhibition of neuron activity noninvasively [125], which could be very appropriate for silencing GCs in the deep layer of the OB or neurons in the OC in the deep brain. More work needs to be done for the development of cell type-specific promotors and animal lines for olfactory neural populations of interest. Another opsin for optogenetic silencing (ST-eGtACR1) has been generated for precise control of neural populations with high fidelity [84], which is potential for in the research of olfactory perception that requires precise temporal and spatial manipulation. In addition, existing optogenetic tools used in the olfactory system, including rhodopsins, chloride, and proton pumps, show some side effects when the reversal potential is higher than the membrane resting potential [126]. The available light activated cAMP and cGMP cyclases are the candidates to address these limitations. However, the application of these novel tools in the olfactory system remains in its infancy. Gao and colleagues activated a cyclic nucleotide-gated channels (olfactory/T537S in oocytes) after photostimulation of a new optogenetic tool (BeCyclOp), which is a guanylyl cyclase rhodopsin from *Blastocladiella emersonii* [127]. Therefore, more work needs to be done in the future to extend the optogenetic toolbox and targeting strategies so that various light sensitive proteins would target various neurons in the olfactory system. Another practical concern is the choice of optical stimulation. When manipulating neurons with light, it is usually required to simultaneously record electrophysiological signals of nearby cells. It is easier to stimulate neurons in slices (in vitro). A laser or LED coupled with an optical fiber or objective is often used to delivery light on slices with defined thickness on a near-uniform manner. An optical fiber is commonly used for in vivo experiments to guide the light and illuminate the surface of the brain (e.g., the surface of the OB or the cortex). However, when it comes to deep brain recording, it is difficult to insert the recording electrodes and the optical fiber to the same site due to lack of visibility. A novel optetrodes, including one optic fiber for light stimulation and four tetrodes for intracellular recordings, has overcome this challenge. In this configuration, it is much easier to manipulation the neurons in the vicinity of the recording site, which has been applied in the recording of the OB neurons. 

The research progress of optogenetics in the olfactory bulb provides ideas for the next direction. First, the mitral cells and the secondary cells receive different OSN signals in the odor information processing in the glomerulus, which indicates the difference in their odor information processing functions, and these two types of cells may be further subdivided into different functional subtypes with the help of optogenetics. Secondly, researchers use optogenetics to link the spatiotemporal activities of the olfactory bulb with olfactory perception and behavior. A meaningful direction is to use this connection to construct biosensors for odor recognition and concentration estimation, which requires the combination of optogenetics and electrophysiology. Furthermore, olfactory bulb interneurons mediate different inhibitory circuits. These OB interneurons are constantly being replaced. Among them, the role of newborn PGCs in odor information processing has not yet been elucidated. Their specific light activation can help explain this problem. In addition, the input from the olfactory bulb indicates that PCx plays an important role in odor characterization, while other areas in the olfactory cortex, such as olfactory tubercle, also receive olfactory input from OB. It is necessary to further determine the performance of olfactory tubercle. In short, optogenetics is expected to solve other important problems in the olfactory system in the future.

## Figures and Tables

**Figure 2 biosensors-11-00309-f002:**
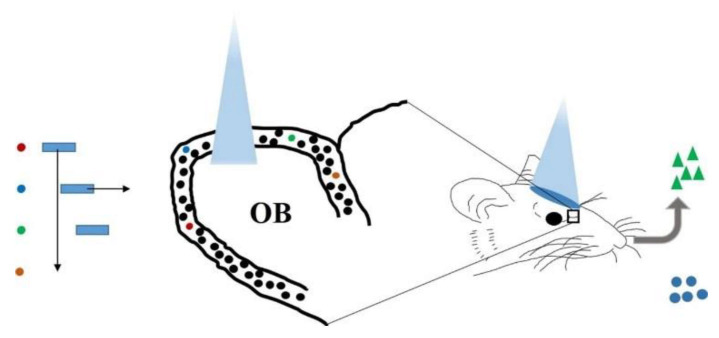
A general strategy for exploring the link between brain activity and olfactory perception using optogenetics. Optogenetics is used to manipulate the spatiotemporal patterns of glomerular or M/T cell activity (such as activation population, latency, duration, etc.), followed by behavioral measurement of animal olfactory discrimination and perceptual limits.

**Figure 3 biosensors-11-00309-f003:**
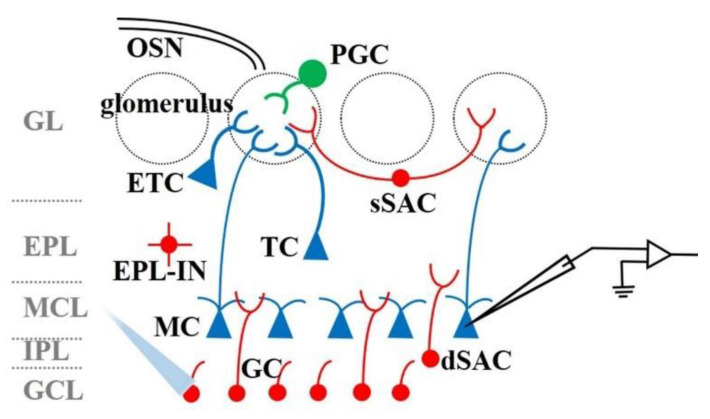
Schematic diagram of the structure of the OB. The method of using optogenetics to study interglomerular interactions is to activate specific interneurons, and then record the electrophysiological signals of M/T cells.

**Figure 6 biosensors-11-00309-f006:**
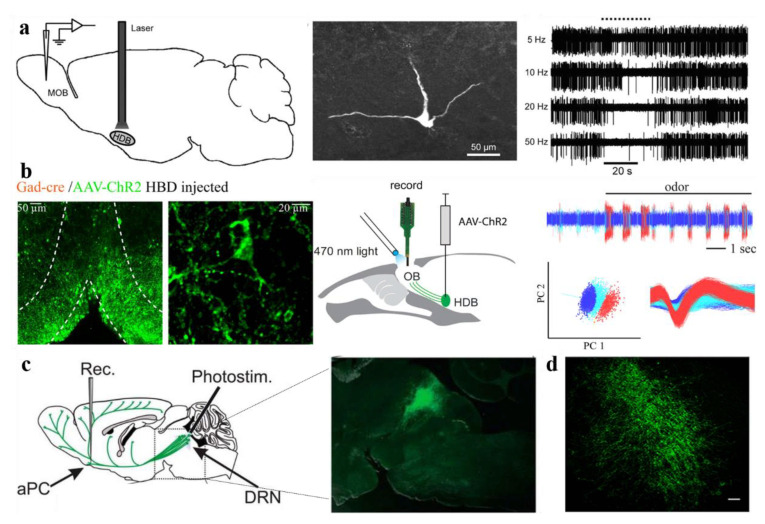
Optogenetic approaches in neuromodulatory inputs to the OB. (**a**) Activating HDB cholinergic neurons inhibits the spontaneous activity of M/T cells [41]. (**b**) Selective targeting of GABAergic inputs from the basal forebrain to the OB. Schematic of experimental approach and data acquisition [63]. (**c**) DRN 5-HT photostimulation suppresses spontaneous aPC activity [65]. (**d**) Optogenetic silencing of norepinephrine axons in the OB does not alter the broadband odorant-elicited change in LFP power [66]. Reproduced with permission from (**a**) [41], Copyright 2012 Society for Neuroscience; (**b**) [63], Copyright 2020 Springer Nature; (**c**) [65], Copyright 2016 Society for Neuroscience; (**d**) [66], Copyright 2018 Ramirez-Gordillo, Ma and Restrepo.

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
