# Peer review of "Olfactory Optogenetics: Light Illuminates the Chemical Sensing Mechanisms of Biological Olfactory Systems"

_biosensors, 2021, doi:10.3390/bios11090309_

Round 1

Reviewer 1 Report

Recently, optogenetics is widely applied in dissecting the olfactory systems. It is good to see a summary of recent discoveries in this area. The authors well summarized the recent advances in olfactory system study.  I would like to recommend the paper to be accepted if the authors could address the several small points below.

My major concerning point is that the perspective part is a bit shallow. I hope that the author could describe a bit more about the desired optogenetic molecular tools and optical devices based on the characters of the olfactory system. For example, there are many other types of optogenetic tools related to cAMP/cGMP and light-induced protein interaction, etc., will those be useful in olfactory system researches? and what kind of optical devices will facilitate the experimental operation? The author already mentioned shortly that " introducing the enhanced tool eNpHR3.0, ST-eGtACR1, and Jaws, etc. may have sparks with olfactory research", this is a good example to go deeper into the perspective part. But what kind of sparks and why should be explained, otherwise it will be too superficial.

Small points,

The two tables should be optimized, especially the "light delivery tool" should be more clear, for example, the authors wrote LED or Fiber optics. They are very blurred. Are the LEDs or the fibers implanted or illumination is performed above. Are the fibers coupled to laser or LED, since the fiber can not generate light by itself.

Line 57, Natromonas pharaonis should be italic.

Line 666, Finally, Finally, to Finally

Figures 1, 4, 5, 6 can be a bit bigger since some texts are too small.

Reviewer 2 Report

The paper needs to be completely revised as it has errors and very hard to read

critically, some of the references are wrongly described.

- [62] used tetrodes and not patch clam and they injected to the GCL and not IPL

- [56] used Pcdh21-Cre mice (Nagai et al., 2005) to restrict starter cells to M/T cells and injected to the M/T cells and not the GL 

Also, the English must be edited. there are many typos and unclear sentences

Reviewer 3 Report

The authors have provided a detailed and comprehensive digest of the findings on sensory inputs, perception of odor, and functional neuronal circuits in the olfactory system that uses optogenetics as a tool. This reviewer finds this review paper well segmented, cohesive, and expressed in a manner that engages the reader. This reviewer, however, thinks that the following improvements need to be made.

  1. Please mention the institution as well, when referring to research led by other groups. Examples include lines 253 and 362.
  2. The quality (text and clarity) of the images needs to be improved. Specifically, fig 4: a,b,d; fig 6: a,c.   
